# Water Reserves for the Environment: A Strategic and Temporal Analysis (2012–2022) for the Implementation of Environmental Flows in Mexico

Sergio A. Salinas-Rodríguez [1,*] and Anuar I. Martínez Pacheco [2]

1    Department of Sustainability Sciences, El Colegio de la Frontera Sur, Villahermosa 86280, Mexico
2    World Wildlife Fund Inc., Mexico City 03100, Mexico; anuarmartinez@wwfmex.org
*    Correspondence: ssalinas@ecosur.mx; Tel.: +52-993-313-6110 (ext. 3464)

**Abstract:** In Mexico, the evaluations of environmental flows are regulated by the Mexican Norm NMX-AA-159-SCFI-2012, and they warrant the establishment of water reserves for the environment. However, the pressure or demand for water use limits the establishment of said reserves because their implementation is generally conditioned to water availability. This research aimed to evaluate the changes through time of the variables that serve as a basis for the implementation strategy by the Mexican government. A geographical information system was built with updated information on water availability, conservation values, and pressures for all basins nationwide. Their desired conservation status was analyzed, and the potential reserves were estimated based on the reference values. The results were examined according to the ranking changes in environmental water reserves enactment feasibility and desired conservation status of Mexican basins, the progress achieved to date, and the potential contribution to the conservation of protected areas and their connectivity if the gaps of reserves were implemented. The outcomes point towards an administrative implementation strategy with positive results despite the growing demand for water use, with a change rate higher than the one for the creation of new protected areas. Currently, basins with low demand and high conservation value have the potential to meet people's and the environment's water needs, and contribute to 86% of the goal set by the present administration without affecting water availability. Finally, reserving water in the priority basins would guarantee the legal protection of the flow regime in 48–50% of the hydrographic network (63,760–66,900 km) in a desired conservation status, 43–49% of wetlands of international importance (48,650–49,600 km$^2$) and other protected areas (128,700–136,500 km$^2$) in 85–89% of the global ecoregions represented in Mexico (780,500–852,200 km$^2$).

**Keywords:** connectivity; basin; ecological importance; environmental objectives; pressure or demand for water use; water allocation

## 1. Introduction

The circulation of water on the planet has been fundamental to the development of civilizations throughout history. The use that societies have given to the water from rivers and wetlands has been the foundation of their well-being. The role played by the environment as a guarantor of ecosystem services is crucial, as these depend on the integrity of ecological processes such as connectivity for water and food provision, and the regulation of both climate and the ecosystems that mitigate the adverse effects of extreme events such as droughts and floods [1–3].

Despite the value of water in nature, aquatic ecosystems are by far the most threatened globally due to overexploitation, modification of flow regimes for productive uses, and pollution resulting in the loss or degradation of habitat and the establishment of exotic and invasive species, along with other emerging threats such as climate change [3–5]. Although these ecosystems cover only 1% of the world's surface, they harbor 10% of all known species [6], yet approximately 83% of the species abundance within them has been lost in

the last 50 years [7]. Even though rivers connect terrestrial and marine environments, only 37% of large rivers maintain high levels of connectivity (conservation status index ≥95%), and only 17% have some form of protection scheme [8,9].

Environmental flows are defined as the quantity, timing, and quality of freshwater flows and levels necessary to sustain aquatic ecosystems which, in turn, support human cultures, economies, sustainable livelihoods, and well-being [2]. They are widely recognized in the specialized literature as a central element for both environmentally sustainable and comprehensive water management, and for achieving related Sustainable Development Goals [2,3,10]. In Mexico, their assessment is guided by the Mexican Norm NMX-AA-159-SCFI-2012 [11], which establishes the procedure for determining the environmental flow in hydrological basins, and has been a cornerstone for environmental water allocation implementation achieved to date to mediate between the needs for productive uses and the conservation of ecological processes and ecosystem services [12,13].

An environmental water reserve (EWR) is an administrative instrument for management grounded on Article 41 of the National Water Law through which a volume is allocated to remain in the environment based on the application of the Mexican Norm NMX-AA-159-SCFI-2012 [12,13]. The environmental flow standard sets a three-level hierarchical framework to conduct assessments, from relatively simple and cheap methods (i.e., "look-up tables" and hydrology-based) to more comprehensive and expensive ones (i.e., holistic ecohydrology-, habitat simulation-, expert panel-, and research driven-based), and provide water reserve outcomes adjusted to a desired conservation state of basins [12,13]. In 2012, the country set the goal of reserving water for the environment in 189 selected basins due to their availability, low usage rates, conservation interest, and other favorable attributes [12–14]. In such a set of basins, a preventive strategy for limiting water abstraction to sustainable rates was piloted to halt the increasing pressure or demand for water use to which basins are subjected [3,15,16]. This strategy aimed to meet the water needs of the present, without compromising future generations' needs [17].

More than ten years after the initiative to allocate water to the environment started, studies have been developed to systematize experiences regarding the performance of methods for the evaluation and the implementation of environmental flows, and the commitment to continue reserving water in the country has been reaffirmed [13,17–21]. Currently, however, no research has been conducted on changes in the key variables for establishing water reserves and implementing the Mexican Norm NMX-AA-159-SCFI-2012, even though it establishes the desired conservation state of basins based on demand for water use and ecological importance at that time [11,22]. Likewise, its effects on public policy goals and on instruments to guarantee long-term environmentally and socially sustainable use have not been analyzed, nor has the potential contribution to the conservation of protected areas, their connectivity, and on the biodiversity dependent on the integrity of flow regimes. Filling this knowledge gap is essential to contribute to the water security public policy agenda.

In the research reported here, the changes over time (2012–2022) in the variables that affect the feasibility of establishing water reserves for ecological protection were evaluated, and the first assessment of environmental flows in the country's hydrological basins for water planning purposes was conducted, in order to identify areas with sufficient availability and those that are deficient. Additionally, the results were examined based on their potential contribution to the conservation of the connectivity of protected areas, dependent ecosystems, and what this means for the global ecoregions of aquatic ecosystems present in Mexico. Understanding the temporal changes in the variables that underlie the strategic framework for evaluating environmental flows will contribute to the update, monitoring, and long-term implementation of public policy goals in a broader context.

## 2. Materials and Methods

The methodological procedure consists of three main stages. The first stage focuses on the collection of updated information, useful for water and protected area management,

and its geographic processing. The second stage encompasses both the fundamentals and the detailed evaluation for prioritizing basins with feasibility as potential water reserves and the environmental objective class (desired conservation state) for environmental flow assessments based on the principles of the Mexican Norm NMX-AA-159-SCFI-2012. Lastly, the third stage incorporates the analysis of changes from 2012 to 2022, progress to date in water allocation to the environment in the country, and the potential scope of results based on three considerations for the conservation and management of dependent ecosystems:

1.  Natural areas such as protected areas, internationally important wetlands (Ramsar sites), and gaps in epicontinental aquatic conservation which are country-scale conservation goals for species, habitats, and vegetation types of freshwater-dependent environments set at 1:1,000,000 resolution based on the optimization of their ecological importance (i.e., risk national or international protection listings), distribution, coverage, species richness, and anthropogenic threats (MARXAN model 1.8.10 was used [23]), officially recognized by the Mexican state [14].
2.  Water resource conservation and management based on public policy for water planning and climate change adaptation [21,24].
3.  The potential contribution to connectivity and representation in global freshwater ecoregions [8,25].

### 2.1. Definitions, Data Sources, and Geographic Processing

A potential water reserve consists of a hydrological basin as a basic management unit, identified as feasible for the development of detailed on-site studies of environmental flow to support its establishment, given favorable characteristics such as water availability, low demand for its use, and conservation interest [12–14,17]. To ensure the temporal analysis under the same criteria, the collection of updated information was based on the following variables: water availability (updated usually every 3–5 years) and stress status (water exploitation index), conservation value (natural protected areas, internationally important wetlands, and gaps in epicontinental aquatic conservation), presence of bans, hydraulic infrastructure, agricultural activities, and population density (Table 1) [12,14]. While the water availability, bans, and reserves are river basins or aquifers that are geographically dependent on specific studies published in the Official Journal of the Federation (i.e., surface and groundwater annual balances at 1:250,000, and environmental flow assessments), the water stress status is calculated based on the corresponding volume for productive uses (i.e., consumptive use) recorded in the Public Registry of Water Rights and published in said studies. Regarding the spatial resolution of the remaining variables, protected areas regardless of the level (international, national, subnational, municipal, or private) are legally tied by decrees published in the corresponding official journal or are property titles-based, and the inhabitant population is census-based.

The source information used to build the database comes from official sources, including the National Water Commission (CONAGUA as in Spanish), the National Commission of Natural Protected Areas (CONANP), and the National Commission for the Knowledge and Use of Biodiversity (CONABIO) under the umbrella of the Secretary of the Environment (SEMARNAT), the National Population Commission (CONAPO), and the National Institute of Statistics and Geography (INEGI). It was obtained in vector format (shapefile) through institutional transparency portals, personal communication [26], and ad hoc calculations. All of the information was aggregated according to hydrological basin, as the basic functional unit for water management and administration on the national geographic continuum [20] (Supplementary Materials).

**Table 1.** Key variables by hydrological basin and information sources for the identification of potential water reserves. Source: Authors' own elaborations based on [12,14].

| Variable | Reasoning | Source |
|---|---|---|
| Water availability. For surface waters, it is the difference in volume that results between the mean annual runoff from the hydrological basin downstream and the volume committed by water usage. For groundwater, it is the difference between the mean annual recharge volume, the natural committed discharge, and the extraction of groundwater. | It is the determining indicator for the creation of reserves with a preventive focus. If there is no water available in the hydrological basin, water cannot be allocated to the environment. | Surface and groundwater mean annual site-specific balances; SEMARNAT and CONAGUA [20,26,27]. |
| Water Stress. The percentage relationship between the mean annual surface runoff, including that generated within the own basin and that coming from upstream sources (or recharge in the case of aquifers), and the volumes extracted for productive water uses, losses due to evaporation, and reservoir level variations. | It is an indicator of the degree of water resource exploitation (i.e., pressure for use). With higher demand, there is increased competition for water and reduced potential for establishing reserves. | Calculated based on water availability studies [20,26,27]. |
| Water bans. A management tool in response to water overexploitation, or in situations of extreme drought (severe scarcity), or in an emergency caused by water pollution, exploitation, use, or utilization. | The existence of water bans is crucial for the establishment of reserves as they provide legal precedent, and are the basis for the reserves with an availability preventive cause. | SEMARNAT and CONAGUA [20,26,27]. |
| Water reserves. A management tool to allocate volumes for domestic or urban–public use, energy generation for public service, or to ensure minimum flows for ecological protection, including the conservation or restoration of vital ecosystems. | Legal foundation for their establishment and an indicator of progress in public policy for the allocation of water to the environment. | Environmental flow site specific assessments; SEMARNAT and CONAGUA [17,20,26,27]. |
| Conservation value. Natural Protected Areas at federal, state, and municipal levels, as well as areas voluntarily designated for conservation, internationally important wetlands (Ramsar sites), and the priorities, gaps, or omissions in the conservation of epicontinental aquatic biodiversity. | The circulation of water in the environment is fundamental for sustaining the ecological functioning of natural areas recognized by the state. Conservation objectives such as species, habitats, and vegetation types are linked to these spaces; therefore, they require the establishment of water reserves for ecological protection. | CONABIO and CONANP [23,28,29]. |
| Irrigation districts and units. | Irrigation farming has a direct effect on land use change and alterations to the natural runoff regime, which is key to establishing the environmental flows that support the reserves. | CONAGUA [30]. |
| Location and volumes of large dams ($\geq$15 m curtain height compared to the maximum level of ordinary flow or $\geq$3 Hm$^3$ capacity). | Demand of water use indicator; it may limit the reserves. | CONAGUA [31]. |
| Total population, density, and growth rate. | Development indicators; the higher the population, density, and growth rate, the larger the demand for water use and lower feasibility for establishing reserves. | CONAPO and INEGI [32–34]. |

In order to guide the development of the global information matrix for the historical analysis, the reference used was the dataset created and managed by the World Wildlife Fund Inc. (WWF) for the National Water Reserves Program, available on the CONABIO web portal [23,28]. The geographic processing was carried out using ArcGIS 10.8 and QGIS 3.28.3, and it involved the integration of variables grounded on their spatial location, using topological intersection methods, or recalculating based on the presence or absence of the incorporated coverages on the resulting layers.

### 2.2. Criteria for Prioritizing Basins for Environmental Water Allocation

The same procedures of the original studies hold public interest criteria, established in the policy instruments. For the prioritization of basins as potential water reserves, an assessment was conducted based on the relative weight assigned to each variable, whether exclusive, positive, or negative according to its interpretation criteria and weighting factors (Table 2).

**Table 2.** Variables, interpretation, criteria, and weighting factors for the feasibility assessment of potential water reserves. NPA: natural protected area (all categories considered: flora and fauna protection area, natural resources protection area, national monument, national park, and biosphere reserve); $B_s$: basin surface; $I_{ds}$: irrigation district surface; $A_s$: aquifer surface; $D_{cv}$: dam capacity volume; $B_{mar}$: basin mean annual runoff (generated within the basin and the one coming from upstream). Source: elaborated by the authors from [12,14].

| Variable | Interpretation | Criterion | Value |
|---|---|---|---|
| Surface water availability (I) | Excluding | Availability < 0 Hm$^3$ | (-.--) |
| Water stress | Excluding | Exploitation $\geq$ 10% | (-.--) |
| Surface water availability (II) | Positive | Volume > 0 Hm$^3$ | 1 |
| Conservation value | Positive | Presence of Ramsar site | 1 |
| | | Presence of a Federal NPA | 1 |
| | | A total of $\geq$34 gaps and omissions of epicontinental aquatic conservation | 1 |
| Bans | Positive | Presence | 1 |
| Irrigation districts | Negative | $I_{ds}/B_s \leq 1\%$ | 0 |
| | | $I_{ds}/B_s \leq 10\%$ | −0.25 |
| | | $I_{ds}/B_s > 10\%$ | −0.5 |
| Dams | Negative | $D_{cv}/B_{mar} \leq 1\%$ | 0 |
| | | $D_{cv}/B_{mar} \leq 10\%$ | −0.25 |
| | | $D_{cv}/B_{mar} > 10\%$ | −0.5 |
| Risk of impact to the basin from groundwater extraction (groundwater stress) | Negative | Low= $A_s/B_s < 100\%$ | 0 |
| | | High= $A_s/B_s < 1\%$ | 0 |
| | | High= $A_s/B_s \leq 10\%$ | −0.25 |
| | | High= $A_s/B_s > 10\%$ | −0.5 |
| Population density | Negative | Density $\leq$ 25 inhab/km$^2$ | 0 |
| | | Density $\leq$ 50 inhab/km$^2$ | −0.25 |
| | | Density > 50 inhab/km$^2$ | −0.5 |

To assess and integrate the level of exploitation of aquifers and their influence on the basin, the treatment initially involved calculating water stress due to groundwater extraction, and subsequent adjusting of the indicator by its risk of impact at the surface level. In general, based on international recommendations grounded on preventive water management practices and some field evidence, groundwater pumping below 10% of the monthly natural baseflow is considered a low risk for ecological protection [35]. For this study, given that surface and groundwater balances are nationwide provided at an annual scale [20,27], the groundwater stress index was simplified to low (<40%) and high ($\geq$40%) (extraction/recharge). However, to ensure the interpretation of this index as preventive for ecological protection, the relative weight assigned to groundwater extraction was basin-adjusted based on the extent of the surface difference, without risk if (a) the low-stressed aquifer surface is the same as the basin's, or (b) high-stressed labeled only if it intersects <1% with the basin's [12,14]. A distinction was made between wetlands of international importance and the gaps and omissions of epicontinental aquatic conservation, and weighted separately, because such spaces are not necessarily given the same legal protection scheme as other protected areas at a federal level (i.e., promoted and managed by a province or subnational entity different than the Federal Natural Protected Areas Commission). The integration algorithm is presented in Equation (1), and the cut-off criteria for the final feasibility classification are as follows: very high = 4.25–5, high = 3.25–4,

medium = 2.25–3, not a candidate or low $\leq 2$, and exclusion of basins without water availability or demand $\geq 10\%$.

$$P = F_{AV} + F_{CI} + F_{BANS} + F_{ID} + F_{INF} + F_{GEI} + F_{PD} \qquad (1)$$

where P is the priority to conduct the environmental flow studies that sustain the feasibility of the declaration of water reserves; $F_{AV}$ is the availability factor (>0 Hm$^3$ and water stress < 10%); $F_{CI}$ is the conservation interest; $F_{BANS}$ is presence of bans; $F_{ID}$ is irrigation districts; $F_{INF}$ is hydraulic infrastructure (dams); and $F_{GEI}$ is groundwater extraction impact; $F_{PD}$ is population density. The temporal analysis was performed by comparing the results of previous analyses from 2011, 2013, and 2016.

The classification of environmental objectives for environmental flow assessments was carried out based on the conceptual principles of the Mexican Norm NMX-AA-159-SCFI-2012. In summary, this standard recognizes that the seasonal and interannual variability of the natural hydrological regime, in terms of attributes such as magnitude, frequency, duration, timing, and rate of change, as well as its components of ordinary and extraordinary flows, are essential for the development of ecological processes, functions, and ecosystem services. It also acknowledges that as water resources are used and the regime is modified, the biological–ecological condition of the ecosystem degrades [13,17–19]. Apart from the various environmental flow assessment methods used for water planning and management or research purposes (e.g., hydrological, ecohydrological, habitat simulation, or holistic approaches) [16,36,37], the environmental flows are assessed based on the assignment of environmental or management objectives. These objectives establish a balance between water uses, resource conservation, and dependent ecosystems [10,12]. The classification of environmental objectives for setting environmental flow recommendations in regulatory instruments serves as the core of public policy in water administration and management. This is because it determines the level of integrity of the environmental flow regime, which in turn underpins the legal framework for protection.

In the Mexican case, an environmental objective represents ecological integrity, defined as the degradation level of an ecosystem caused by human activities that triggered the loss or transformation of its structural and functional characteristics [11]. These objectives are established, as a function of the pressure for water usage or demand in the basin and its ecological importance, in a combination of classes (Figure 1), where "A" implies a very good desired conservation state, "B" a good one, "C" a moderate state, and "D" a deficient state [12,13,17,18,38]. Thus, the model incorporates fundamental ecohydrological principles that emphasize the system's dependence on the natural hydrological regime, as well as the ecological consequences of its alteration [10,39,40].

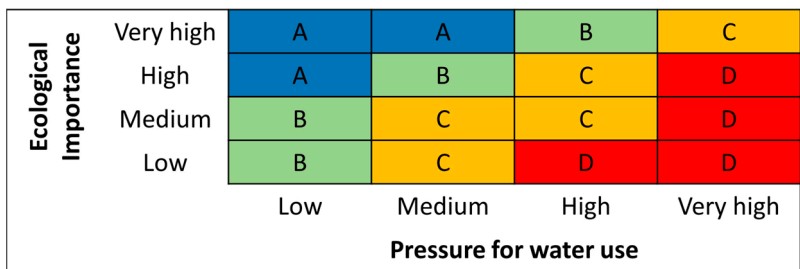

**Figure 1.** Matrix for the classification of environmental objectives. Source: Mexican Norm NMX-AA-159-SCFI-2012.

The classification of pressure for water use was based on the percentage relationship between availability and the volumes extracted for productive uses, losses due to evaporation, and reservoir level variations (low $\leq 10\%$, medium $\geq 11\%$, high $\geq 40\%$, and very high $\geq 80\%$ = high). It is important to note that for the existing water reserves for ecological protection, the volumes designated for this purpose were reintegrated into availability to conduct the assessment under the same conditions as the initial evaluation. Regarding

the ecological importance, consideration was given to the presence of natural protected areas of any level (federal, state, municipal, or private), internationally important wetlands (Ramsar sites), and gaps or omissions in epicontinental aquatic conservation. Concerning the latter, at least 34 gaps and omissions were considered. The value was obtained from the central range theory of distributions to identify the sites with the highest concentration according to the frequency in the last quartile. Final classification of ecological importance was given based on the following criteria: very high if all three conditions are met = natural protected areas, Ramsar site, and ≥ 34 conservation gaps; high if at least two conditions are met; medium if at least one condition is met; and low if there are no protected natural areas, Ramsar site, and ≥ 34 conservation gaps [12,22]. Lastly, for the temporal analysis, the results were compared with the environmental objectives published in the Mexican Norm NMX-AA-159-SCFI-2012 and the updated information from availability balances, natural protected areas, and internationally important wetlands as of 2016 [13,41].

*2.3. Temporal Analysis: The Path to the Current State and Assessment of Achievements*

2.3.1. Conservation and Management of Protected Areas

Despite the conceptual recognition of the importance of freshwater circulation in the landscape for the conservation of natural protected areas and internationally important wetlands (Ramsar sites), currently, only one site includes in its management program the role of the environmental flow in its conservation: the Marismas Nacionales Biosphere Reserve in Nayarit [18,42]. Hence, the temporal analysis of results included the extension of these natural areas located in the basins of potential water reserves, the area benefiting from conservation gaps and omissions, and progress to date in the legal protection of the flow regime.

2.3.2. Water Reserves for Water Planning Based on Reference Values

The conservation and management of water resources are fundamental for the protection of ecological processes and functions as well as ensuring the provision of ecosystem services on which human rights depend, such as the right to a healthy environment and access to water in a sustainable and timely manner. Therefore, the protection of environmental flows is a goal of the National Water Program 2020–2024 and the Special Climate Change Program 2021–2024 [21,24]. In this regard, the difference between the number of basins with existing reserves and potential reserves was examined, along with the opportunity it presents to advance in compliance with the mentioned public policy instruments.

In line with the above, in this contribution we provide the first comprehensive country-scale assessment of water reserve volumes for ecological protection in all the Mexican basins that currently do not have such a scheme. Although similar environmental flow estimations are currently available and could inform implementation progress against water scarcity, they are global-scale-model-derived, display significant gaps or inconsistencies in arid, and semiarid climates, and limit confidence in their relevance in countries like Mexico [43–45]. This assessment is based on systematic analyses of outcomes reported in scientific literature that have demonstrated consistency between hydrological, ecohydrological, and holistic methods generated in nearly 300 case studies in Mexico [13,17–19].

The water reserve volumes for the country's basins were determined based on their assigned environmental objectives according to their ecological significance and current demand for water use, except in 266 basins where water reserves were already enacted; in these cases, reserved volumes were used [17,46,47]. The reference values from the ecohydrological method were based on the frequency of occurrence and seasonal and interannual variability of hydrological regime components, as recognized in the NMX-AA-159-SCFI-2012.

Among the available reference value options, this study used water reserve volumes based on the central distribution range of percentages of the mean annual runoff for each environmental objective class of each analysis unit. For class "A", it was 59% of the runoff generated within the basin plus that from upstream (runoff within the basin plus upstream

runoff in availability balances), "B" was 44%, "C" was 35%, and "D" was 26% [17]. These reference values are expressed as percentages of the mean annual runoff and are recognized in the international literature solely for water planning purposes, even when the source of information comes from monthly or annual-scale precipitation runoff models, as is the case in Mexico. Thus, for the purpose of this research, the approximation is appropriate (look-up tables, [16–19,37]). Water reserve calculations were performed using the global database in Microsoft Excel. Basins where water is available in sufficient quantity to ensure environmental flow implementation (surplus) were distinguished from those in deficit by subtracting the ecological reserve based on the reference values to the current availability, and displayed on the map.

2.3.3. Potential Contribution to the Conservation of the Connectivity of Aquatic Ecosystems

The physical–spatial dimensions of connectivity, including longitudinal connectivity between the main river channels and their tributaries along the basin from headwaters to the mouth, lateral connectivity between rivers and adjacent wetlands in floodplains, and vertical connectivity through the interaction of surface, subsurface, and groundwater, are naturally regulated by the temporal dimension [48–53]. Protecting the flow regime is key for conserving and managing the ecological integrity of aquatic ecosystems [39,54,55]. The implementation of environmental flows is a top priority step in global action and emergency plans to reverse or at least halt the degradation of these ecosystems [2,3]. Therefore, legal protection for free-flowing rivers, that is to say, free of infrastructure that alters the flow and flood regime, is a crucial move [8,9,17,53,56,57].

To analyze Mexico's potential contribution to the conservation of connectivity, we examined the length of its free-flowing rivers from headwaters to the mouth and the segments in good conservation status (conservation status index, CSI > 95% [8]) in basins with water reserves (baseline), the potential water reserves from this evaluation, and those with an environmental objective class "A". To obtain the representation of the results, information on free-flowing rivers worldwide corresponding to the Mexican territory was extracted from the free-flowing rivers map server (Hydrolab, https://hydrolab.io/ffr/#3 /25.90/15.79/FFR-CNT-NME-CNN-LKE). From the national hydrographic network, we filtered the connected segments of free-flowing rivers and those in good conservation status with impacts greater than 5% according to the indicator created by the original source (fields INC and CSI_FF2 in the original attribute table [58]). Finally, we also examined the representation of the potential water reserves from this evaluation and those with an environmental objective class "A" in the context of the protected areas, wetlands of international importance, and freshwater ecoregions [25].

## 3. Results and Discussion

### 3.1. Feasibility Status of Water Reseves

The feasibility rankings of river basis range from medium to very high priority for conducting environmental flow assessments because they possess favorable conditions of water availability, low demand, and conservation interest (Equation (1) outcome $\geq 2.25$), to non-eligible ($\leq 2$), or were excluded due to lack of water availability and/or present and stressed condition ($\geq 10\%$ water exploitation index). Unlike the previous analyses, the administrative division of hydrological basins now stands at 757, with 29 more management units than in 2011 (Table 3). Despite this increase, the overall rate of change is noteworthy, and, particularly, the loss of basins identified as potential water reserves (189 in 2011 vs. 146 in 2022, a decrease of 23%). Except for medium feasibility, which has remained relatively stable over time (~116), the rest of the prioritization categories have been affected.

**Table 3.** Number of basins by their level of feasibility as potential water reserves for ecological protection through time (2011–2022).

| Feasibility | 2011 | 2013 | 2016 | 2022 |
|---|---|---|---|---|
| Very High | 19 | 19 | 3 | 4 |
| High | 54 | 48 | 36 | 26 |
| Medium | 116 | 116 | 108 | 116 |
| Potential water reserve | 189 | 183 | 147 | 146 |
| Non eligible | 268 | 276 | 286 | 301 |
| Analyzed basins | 457 | 459 | 433 | 447 |
| Excluded basins | 271 | 273 | 298 | 310 |
| Total basins | 728 | 732 | 731 | 757 |

In 2011, 54 basins with high feasibility were identified, while in 2022, there were only 26 remaining, roughly half. In 2011, 19 basins had very high feasibility, but in 2022, only 4 met the desired conditions. This can be explained by two reasons: the first is related to the number of basins that met all the assessment conditions (457 vs. 447). The second reason, which is related to the first, is due to the increase in excluded basins from the lack of mean annual surface water availability or an increase in water stress (271 vs. 310). This finding demonstrates that water use for productive uses continues to be environmentally and socially unsustainable despite the existing public policies and regulations during the analyzed period.

In terms of their geographical distribution, there is a significant number of potential water reserves in the Baja California (Northwest) and Yucatán (East) peninsulas. The first, despite the limited natural water available, this indicates that population growth remains low. The second reason is due to the abundant surface and groundwater available, with high conservation interest in both cases (Figure 2). The number of basins identified on the Pacific Ocean side stands out, subject to seasonal hydrological variability that distinguishes it significantly from those draining into the Gulf of Mexico side. Finally, it is worth noting that in the northern part of the country, there is still water available for the environment. In addition to the aforementioned potential reserves in the Baja California peninsula, there are also management units which meet the established criteria, located in the northwestern states of the hydrological regions 25 San Fernando–Soto La Marina and 34 Cuencas Centrales del Norte.

*3.2. Effect of the Change on Protected Areas and Recognized Conservation Gaps*

Currently, Mexico has a system of 187 federal protected areas covering a total of 909,673 km$^2$, encompassing terrestrial ecosystems (173,818 km$^2$ of continental surface, islands, and epicontinental water bodies), exclusive marine areas (440,433 km$^2$), and mixed environments (295,422 km$^2$ in coastal or terrestrial-marine areas) [59]. This evaluation of potential water reserves extends over 76,723 km$^2$, with direct or indirect benefits to protected areas in case of enacting the environmental flows that sustain them (Table 4). This represents a difference of 9408 km$^2$ compared to the initial report (11% rate of change during the analysis period) [14].

As for internationally important wetlands, there are currently 145 Ramsar sites covering an area of 87,274 km$^2$ [60]. Since ecological processes in wetlands are dependent on flood regimes, 37,379 km$^2$ are identified with direct benefits. This indicator shows an additional 8367 km$^2$ compared to the previously reported figures (29% rate of change) [14].

To fully appreciate the potential benefits of protecting environmental flows through the identified potential reserves, it is necessary to consider them as part of the surface of the territory through which water flows, particularly in areas that currently lack legal protection instruments, such as protected areas and recognized gaps and omissions in conservation. In this regard, the global set of 146 potential water reserves implies a drainage area of 411,980 km$^2$. Although this represents a −8% rate of change compared to what was reported previously [14], 90,438 km$^2$ of the area corresponds to federal protected

natural areas and internationally important wetlands. The remaining 321,542 km² contains gaps and omissions in conservation (128,408 km²), and 193,134 km² are identified as new areas of exclusively hydrological protection not covered by any current instrument. This represents a 146% increase compared to the potential water reserves of 2011. This positive difference is due to the increase of 29 hydrological basins from one period to another, with the southeastern peninsula region being a notable addition.

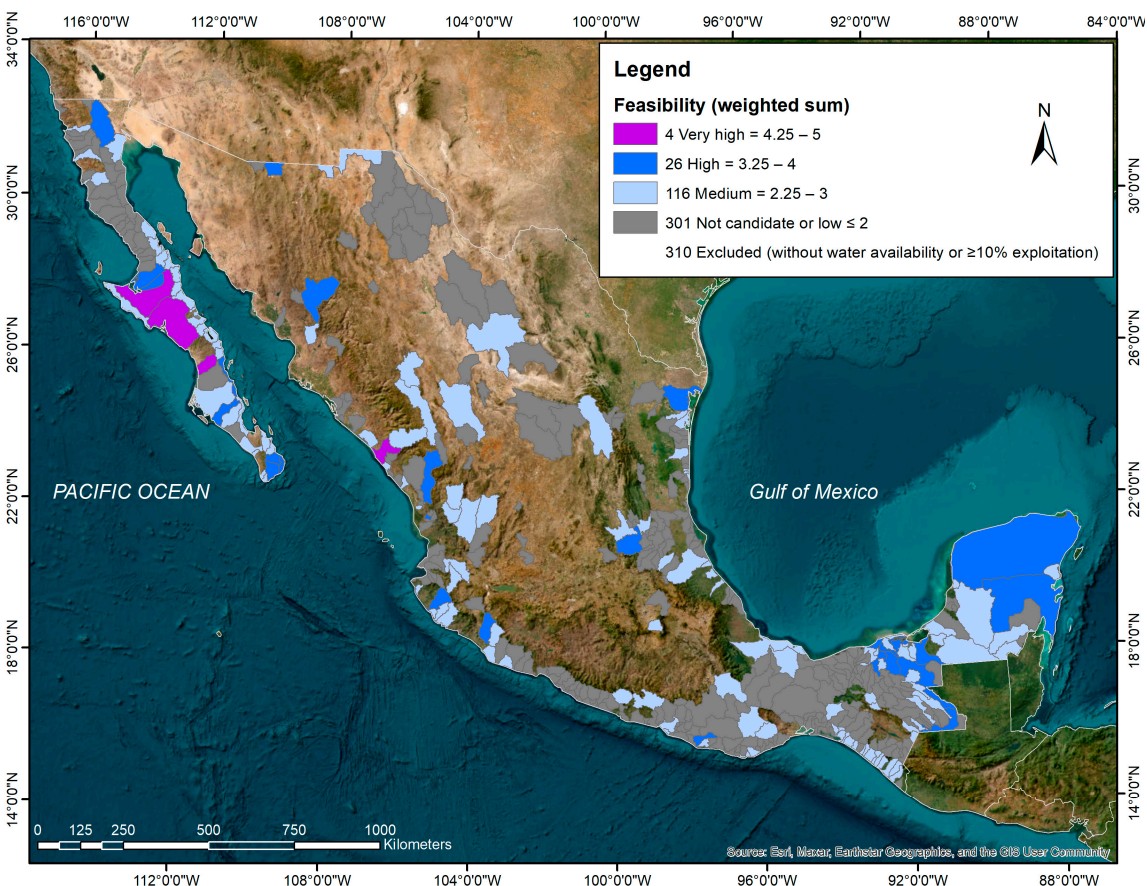

**Figure 2.** Geographic distribution of basins identified as potential water reserves.

**Table 4.** Extensions in square kilometers of potential water reserves, natural protected areas, internationally important wetlands, and gaps and omissions in the conservation of epicontinental aquatic ecosystems.

| Feasibility | Very High | High | Medium | Total |
|---|---|---|---|---|
| Potential water reserve | 26,241 | 144,450 | 241,289 | 411,980 |
| Natural Protected Areas | 15,431 | 23,718 | 37,575 | 76,723 |
| Flora and Fauna Protection Area | 1366 | 8603 | 8527 | 18,496 |
| Natural Resources Protection Area | 0 | 16 | 4310 | 4326 |
| National Monument | 0 | 27 | 42 | 69 |
| National Park | 0 | 194 | 443 | 637 |
| Biosphere Reserve | 14,064 | 14,875 | 24,241 | 53,179 |
| Sanctuary | 0 | 4 | 11 | 15 |
| Ramsar Sites | 7387 | 18,525 | 11,467 | 37,379 |
| Gaps and omissions (class) | 8953 | 38,400 | 81,055 | 128,408 |
| Extreme | 3899 | 9367 | 23,287 | 36,553 |
| High | 3112 | 12,203 | 16,643 | 31,957 |
| Medium | 1942 | 16,830 | 41,125 | 59,897 |

### 3.3. Environmental Objectives for Environmental Flow Assessments

In general, the classification of environmental objectives for hydrological basins follows the same trend as the identification of potential water reserves. Regarding water pressure or demand of use, the temporal analysis reveals increases with rates of change ranging from 5% to 21% between the low and very high classifications, respectively (Table 5). On the contrary, and despite the addition of 25 management units for the analysis period, the ecological importance in the basins reflects an opposing relationship, albeit to a lesser extent (1–12%). Given the changes in both pressure for water use and ecological importance, the classification of environmental objectives has also been impacted; in general, the most significant changes are found in the central categories (B–C). For the environmental objective class "D", representing a desired conservation state labeled as deficient, the number of basins has increased from 156 to 220 (41% rate of change), while for class "C," which represents a desired medium ecological state, the change decreased from 304 to 105 basins (−190%). On the other hand, the change for class "B" increased from 82 to 292 basins (256% with a desired good conservation state), and for class "A," it decreased from 190 to 140 basins (26%).

**Table 5.** Numbers of basins for the evaluation of environmental flows in each level of pressure for water use, ecological importance, and classification of environmental objectives through time.

| Description | 2012 | 2016 | 2022 |
|---|---|---|---|
| Pressure for water use | | | |
| Low | 423 | 387 | 399 |
| Medium | 68 | 64 | 76 |
| High | 34 | 35 | 33 |
| Very High | 207 | 245 | 249 |
| Ecological importance | | | |
| Low | 174 | 182 | 170 |
| Medium | 262 | 252 | 292 |
| High | 230 | 227 | 236 |
| Very High | 66 | 70 | 59 |
| Environmental objective—Desired conservation status | | | |
| D—Deficient | 156 | 144 | 219 |
| C—Medium | 304 | 276 | 106 |
| B—Good | 82 | 93 | 292 |
| A—Very good | 190 | 218 | 140 |

Geographically, basins with environmental objectives of class "A" and "B" are mostly distributed in the two peninsulas, in coastal basins along both the Pacific Ocean and Gulf of Mexico, and with a small representation in the northwestern region, as well as in the central northern basins (Figure 3). In contrast, basins with environmental objectives of class "C" and "D" are concentrated in the northernmost region and in central-southern Mexico.

### 3.4. Gaps in Water Reserves for Management Planning

Out of the 757 basins in the country, 266 have existing water reserves. Among the remaining 491 basins, the use of reference values for reserve volumes for water planning reveals that 323 basins are suitable for establishing new reserves to meet national goals [21,24]. However, 168 of these basins are in deficit, meaning they lack sufficient availability (Table 6; Figure 4). As previously reported, these water availability deficient basins would require detailed environmental flow studies for flow regime reconstruction, adjustments to environmental objectives, and implementation strategies to involve the recovery of consumptive water volumes [12,17].

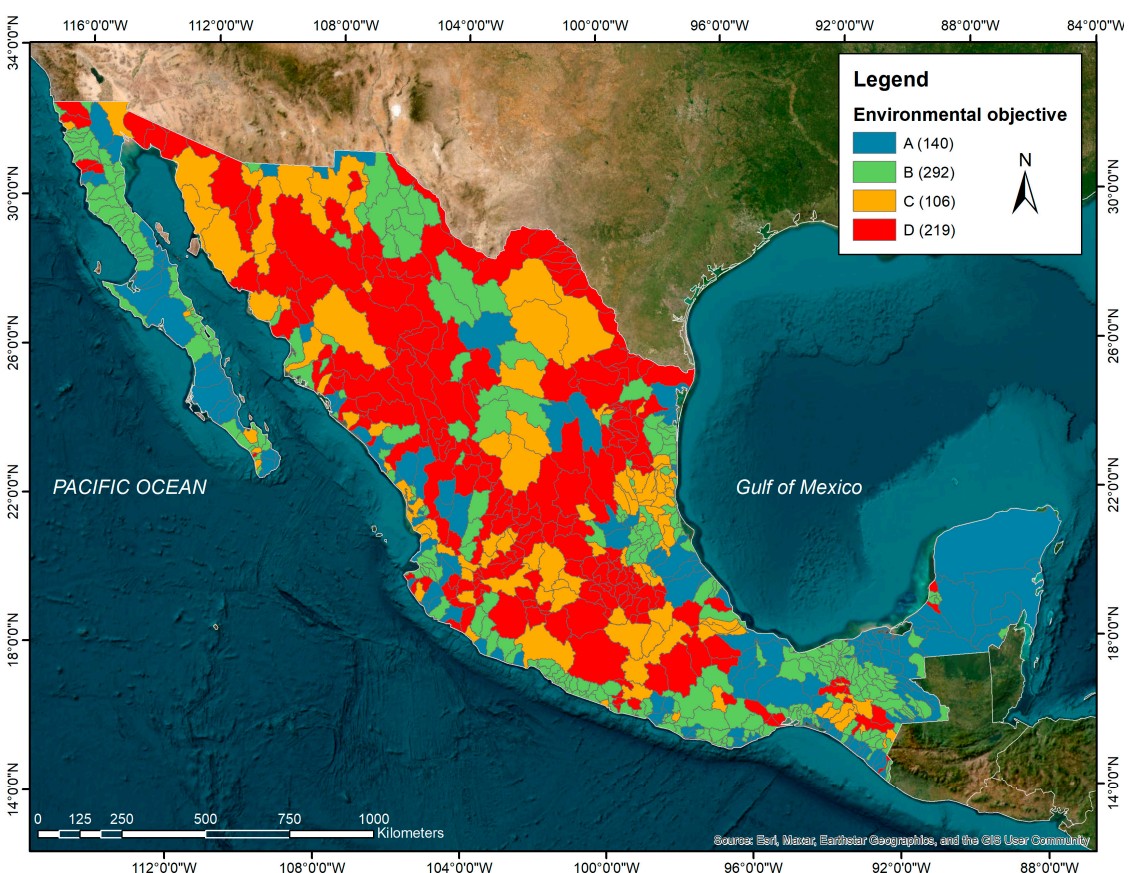

**Figure 3.** Geographic distribution of the environmental objectives' classes by hydrological basin.

**Table 6.** Numbers of basins by environmental objective, with and without available water volume, for the allocation of ecological protection reserves based on the use of reference values for water planning.

| Environmental Objective—Desired Conservation Status | Surplus | Deficit | Total |
|---|---|---|---|
| D—Deficient | 87 | 132 | 219 |
| C—Medium | 80 | 26 | 106 |
| B—Good | 285 | 7 | 292 |
| A—Very good | 137 | 3 | 140 |
| Total | 589 | 168 | 757 |

Regarding the relevance of water reserve volumes for water planning based on reference values, the proportion of basins with a surplus of water availability is concentrated in those with environmental objectives classes "A" and "B" (98% for each class), with the lowest number of outliers (11–12%) (Figure 5). Conversely, in deficit basins, both environmental objectives classes "C" and "D" experience the highest proportions of both conflict (75% and 45%, respectively) and outliers (25% and 29%, respectively).

In general, these results are consistent with the expected as previously reported, both in the method's proof of concept as well as in case studies that include desktop assessments (ecohydrological methods) and others supported by holistic approaches [13,17–19,61,62]. However, the reserve volumes should be taken with caution due to their temporal resolution-related uncertainty, particularly in the desired conservation states of "medium" and "deficient", and further considerations should be taken into account.

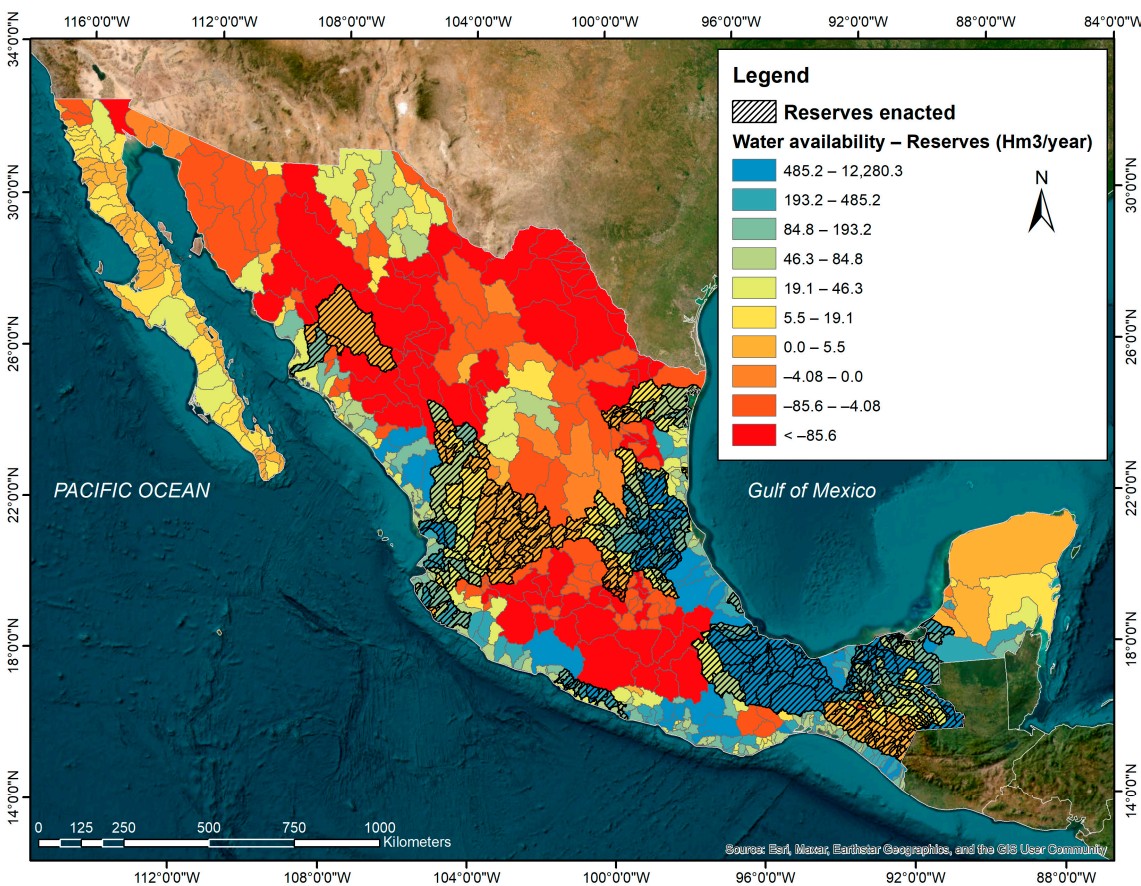

**Figure 4.** Geographical locations of the country's basins with current water reserves, and the surplus and deficit of water availability for the establishment of new reserves (volume of current water availability minus reserves).

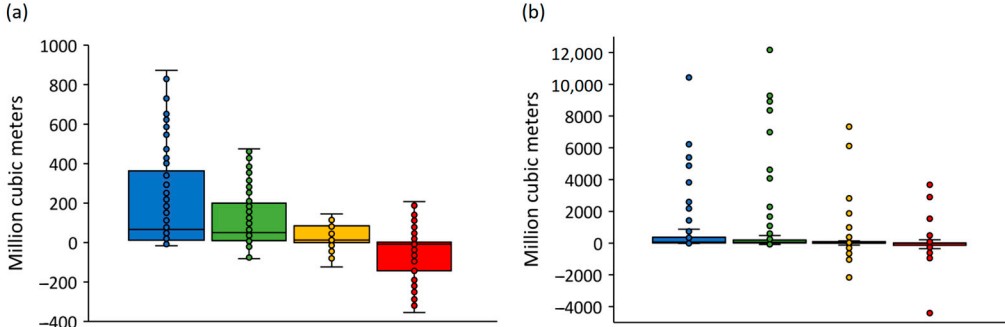

**Figure 5.** Exploratory analysis of water availability volumes in million cubic meters based on reference values for ecological reserve by class of environmental flow objectives (A = blue, B = green, C = yellow, D = red), with focus on (**a**) central frequency distribution values within upper and lower limits (quartile 3 and 1 ± 1.5 times the interquartile range, respectively) and (**b**) outliers.

First, the water availability in the basin is mean annual-based and the extremes are not adequately accounted for (i.e., very dry and wet seasonal and yearly conditions). Although carried out at a country and global scale, annual-based water availability and environmental reserves tend to be a regular practice for water planning; there is evidence of inconsistency at finer scales [43–45]; drought cycles may affect environmental water reserve previsions provided by the reference values. Furthermore, another limitation to be considered is that the reference values were applied without differentiation between the type of dominant stream in the basins (e.g., perennial, intermittent, or ephemeral), since

it is not possible to determine this from the availability balances [17–19]. To increase the certainty in these regards, bottom-up detailed methods are recommended, such as holistic ecohydrology-based, habitat simulation, expert panel, or research-driven methods.

*3.5. Strategic Contributions of the Reserves to Public Policy and Potential Gains for the Conservation of Aquatic Ecosystem Connectivity*

Although the reduction in feasible potential water reserves from 2011 to 2022 may have had an impact on the goal of establishing 189 reserves for ecological protection, currently, 266 have been established and remain in place. From the current set of basins with reserves, only 27 are identified as potential, so there are still 119 possible new establishments in accordance with the national goals [21,24]. As previously reported, the total set of established water reserves exceeded the goal due to management opportunities arising from hydrological connectivity between a select group of units and their contributing upstream basins [17,63]. To ensure the required volumes in the target reserves, it was necessary to ensure connectivity through availability balances, and consequently, reserve the water from the generating or contributing basins.

Unlike previous reports [17], the adjustment of the contribution of current reserves to the protection of flow regimes on the drainage network is around 443,000 km$^2$, providing legal certainty to ensure connectivity, at least in good conservation condition, along ~41,600 km or 31% of the national hydrographic network, ~12,200 km$^2$ in 39 Ramsar wetlands of international importance, ~59,850 km$^2$ in 54 federal natural protected areas, as well as ~442,700 km$^2$ with representation of 70% of the global aquatic ecosystem ecoregions present in Mexico (Table 7). In terms of biodiversity, this benefits at least ~240 protected species, ~180 of which are freshwater dependent (~80 protected) [38]. Moreover, according to a recent report from the International Union for Conservation of Nature (IUCN), this water protection scheme has the potential to improve the conservation status of 450 fish species in Mexico [64]. Based on the present evaluation of potential water reserves and basins with environmental objective class "A", the gain for the conservation of aquatic ecosystem connectivity could increase to around 63,760–66,900 km of rivers with at least a good state of conservation in their connectivity (48–50% of the national hydrographic network), 48,650–49,600 km$^2$ in 62–66 wetlands (43–46%), 128,700–136,500 km$^2$ in 84–92 natural protected areas (45–49%), and 780,500–852,200 km$^2$ with representation of 85–89% of the global freshwater ecoregions present in the country.

The experience gained in the allocation of water for the environment over the last decade is revealing for its scope and is acknowledged in scientific literature [3,9,16,56,57,63,64]. The results have the potential to be leveraged through public policy for water resources management, conservation, and addressing climate change. Currently, the federal government has the goal of increasing the number of basins with ecological reserves to 448, which means an addition of 182 to the existing ones, along with the development of complementing regulations to ensure the flow regime timing implementation [21,24,65]. Focusing the strategy on the remaining set of 119 basins with potential water reserves and the hydrologically connected management units would provide certainty regarding the current water availability, its low usage demand rates, and conservation interest, contributing at least 86% (385 out of the 448 reserves) towards achieving the goals of the National Water Program 2020–2024 and the Special Climate Change Program 2021–2024.

Such country-scale strategic environmental water implementation is aligned to the state-of-the-art calls for protecting and restoring habitats to mitigate the ongoing loss of freshwater biodiversity [2,3,56,57]. It builds on the commitment made by the government of Mexico, together with those from Colombia, Congo, Ecuador, Gabon, and Zambia at the United Nations (UN) 2023 Water Conference for the Freshwater Challenge (https://www.gob.mx/sre/prensa/mexico-joins-the-freshwater-challenge-to-restore-rivers-and-wetlands-and-address-water-scarcity?idiom=en) that aims to collectively restore 300,000 km of rivers and 350 million hectares of wetlands and address water scarcity. Furthermore, it is consistent with Target 3 of the Kunming-Montreal Global Biodiversity

Framework, "ensure and enable that by 2030 at least 30 percent of terrestrial, inland water, and of coastal and marine areas are effectively conserved and managed through ecologically representative, well-connected and equitably governed systems of protected areas and other effective area-based conservation measures [...]", recently adopted by the Convention on Biological Diversity.

**Table 7.** Strategic and potential contributions of the water reserves and numbers of basins with environmental objective class "A" for environmental flow assessments, according to NMX-AA-159-SCFI-2012 on conservation objects dependent on the flow regime.

| Conservation Objects Dependent on the Flow Regime | Current Reserves (266 Baseline) | Potential Reserves (146) | Basins with a Class "A" Environmental Objective |
|---|---|---|---|
| Federal Natural Protected Areas (km$^2$) | 59,852 | 76,723 | 68,841 |
| Wetlands of international importance (km$^2$) | 12,217 | 37,379 | 36,438 |
| Ecohydrological connectivity of free-flowing rivers at least in good conservation state (km) | 41,632 | 25,306 | 22,162 |
| Free-flowing rivers from source to mouth | 37,974 | 23,377 | 20,668 |
| Good conservation status | 3657 | 1930 | 1494 |
| Freshwater ecoregions (km$^2$) | 442,742 | 409,483 | 337,772 |
| Ameca–Manantlan | 23,632 | 16,292 | 11,939 |
| Chiapas–Fonseca | 455 | 14,762 | 8083 |
| Coatzacoalcos | 19,392 | 65 | 65 |
| Colorado | 0 | 6566 | 6566 |
| Cuatro Cienegas | 0 | 0 | 0 |
| Gila | 0 | 1492 | 1492 |
| Grijalva–Usumacinta | 69,325 | 46,010 | 44,030 |
| Guzman–Samalayuca | 39 | 5318 | 5318 |
| Lerma–Chapala | 4621 | 688 | 305 |
| Llanos El Salado | 8831 | 5231 | 5231 |
| Lower Rio Grande–Bravo | 23,469 | 7430 | 7240 |
| Mayran–Viesca | 16,741 | 44,297 | 21,505 |
| Panuco | 73,710 | 17,590 | 26,647 |
| Papaloapan | 50,337 | 15,691 | 10,732 |
| Quintana Roo–Motagua | 0 | 29,801 | 29,801 |
| Rio Balsas | 1706 | 2134 | 1296 |
| Rio Conchos | 743 | 259 | 0 |
| Rio Salado | 0 | 0 | 0 |
| Rio San Juan (Mexico) | 304 | 53 | 53 |
| Rio Santiago | 84,418 | 16,028 | 14,080 |
| Sierra Madre of the South | 8149 | 9372 | 4134 |
| Sinaloa | 34,175 | 17,112 | 13,428 |
| Sonora | 70 | 7888 | 319 |
| Southern California Coastal–Baja California | 0 | 70,311 | 50,415 |
| Upper Rio Grande–Bravo | 22,627 | 11,362 | 0 |
| Upper Usumacinta | 0 | 63,731 | 11,362 |
| Yucatan | 0 | 0 | 63,731 |

## 4. Conclusions and Recommendations

The results of this research show that over the last decade, the form and geospatial distribution of pressure or demand for water use in Mexico's basins has been advancing at a higher rate compared to the creation of new protected areas. Although both aspects are crucial for achieving a balance between water use and conservation, water demand is the most determining factor reported in international literature and, therefore, limits the establishment of water reserves for ecological protection. It is an indicator of the level of over-exploitation of resources whose use alters the regime and pollutes water, leading to the loss or degradation of habitat (e.g., connectivity) and of species living in aquatic ecosystems.

The strategy adopted in Mexico to proactively implement environmental flows through water allocation for ecological protection, in basins with low water usage rates and significant conservation interest, has been successful according to performance indicators established in public policy instruments back in 2012 (266 reserves, 77 more than originally targeted) and recognized by specialized literature. However, this was due to the need—and the water management opportunity—to ensure the connectivity of environmental flows in a limited number of basins through all their contributing units located upstream. Currently, while reserves have been guaranteed in only 27 of the initially identified potential basins, there are still 119 management units that meet the conditions of low water usage rates and significant conservation interest; it is recommended to allocate environmental water in such basins. This number could contribute to approximately 86% of the goal set in national programs by 2024, with the certainty that they have water availability, low demand, and conservation interest for the protection of environmental flows, without considering the need to also ensure those of the contributing basins upstream. This has proven to have positive implications from water management.

The evaluation of environmental flows based on reference values for water planning shows that up to 78% of the country's basins have sufficient and potential availability to manage their administrative implementation through the water reserves legal figure, regardless of their environmental objective class. However, given the associated uncertainty in basins with different desired conservation states, as well as the hydrological nature of the main watercourses (ephemeral, intermittent, or perennial), it is recommended to design differentiated strategies for their implementation. The strategies should be focused on enacting environmental reserves grounded on holistic ecohydrology-based studies in basins with sufficient availability and ecological importance, and developing the complementary regulation to secure timely implementation (i.e., conservation and management plans). The recommended strategy in basins without enough water availability should be focused on the reconstruction of environmental flow regimes, and grounded on holistic research-driven studies to support the corresponding restoration plans.

Finally, in light of our findings, current strategic environmental water allocation in the Mexican basins with certainty of low demand of use and high conservation importance with at least a good state of conservation in their connectivity will provide multiple benefits in the long term. From a management perspective, by enacting the flow regime protection in such basins, water abstractions will be preventively limited for up to 50 years to sustainable rates in around half of the country's river network, protected areas, and wetlands of international importance, and avoid conflicts between people and nature in this increasingly scarce resource. Such implementation would mean an unprecedented country-scale measure to halt biodiversity loss in aquatic ecosystems. Moreover, it will bring Mexico closer to fulfilling the commitments made at the UN 2023 Water Conference and the Kunming-Montreal Global Biodiversity Framework, recently adopted by the parties to the Convention on Biological Diversity.

**Supplementary Materials:** The database for the current national assessment on the potential reserves (ranking), desired conservation status of the flow regime (environmental objective class), and environmental water reserve (reference values-based) per river basin are available online at https://www.mdpi.com/article/10.3390/d16030190/s1.

**Author Contributions:** Conceptualization, methodology, formal analysis, investigation, project administration, supervision, visualization, writing—original draft preparation, S.A.S.-R.; validation, resources, data curation, funding acquisition, writing—review and editing, S.A.S.-R. and A.I.M.P. All authors have read and agreed to the published version of the manuscript.

**Funding:** This research was funded by El Colegio de la Frontera Sur (ECOSUR) and the World Wildlife Fund Inc. (WWF Mexico), grant number ECOSUR/CONVENIO/E/030/VIH/2022. Furthermore, it supports the National Research and Advocacy Project "Ecohydrology for the sustainability and governance of water and basins for the common good", funded by the Mexican National Council of

Humanities, Sciences and Technologies and operated by the Water Reserves Monitoring Network, grant number 318956. The APC was funded WWF Mexico.

**Institutional Review Board Statement:** Not applicable.

**Data Availability Statement:** The data presented in this study are available in this published paper. Shapefiles are available upon request to the corresponding author.

**Acknowledgments:** We are thankful to Sofía Álvarez Castrejón for her support on the manuscript translation and style proofreading.

**Conflicts of Interest:** The authors declare no conflicts of interest.

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
