# Peer review of "Water Reserves for the Environment: A Strategic and Temporal Analysis (2012–2022) for the Implementation of Environmental Flows in Mexico"

_diversity, doi:10.3390/d16030190_

Round 1
Reviewer 1 Report
Comments and Suggestions for Authors
This paper addresses establishment of additional watershed scale water reserves for Mexico using multiple parameters to rank potential sites. This sounds like important work for addressing pressures on water resources and for the ecosystem services associated with water.
I think this paper has potential for strong impact, not only for Mexico but as a template for what could be evaluated in other countries. However, its very difficult for me to understand what are the parameters and how they are being ranked. I don’t know if it’s the translation to English that has made this confusing or that the authors have such a deep background in their topic that they forget to explain fully to readers who are seeing the topic for the first time.
I will share a few examples of where I was confused –
- Abstract: The outcomes point towards an administrative implementation strategy with positive 17
outcomes despite the growing pressure for water use, with a change rate higher than the one for the 18
creation of new protected areas.
What is the ‘change rate’? The prior sentence doesn’t mention ‘change rate’ but talked about
historical, current, and potential contribution to the conservation of natural protected areas and their 16
connectivity
This type of vagueness is challenging for a reader.
-Table 1 starts to describe the variables, and the sources of information – many of which are CONAGUA and CONBIO, plus Mexican Norm. For those of us who aren’t familiar with these databases, descriptions of how they were developed, when and the spatial resolution of the data as well as the uncertainty of the data and database. However, I really question that all these data are available at the detail described and accurate for each basin. That isn’t to say that the analysis of change over time isn’t valuable, but you don’t provide enough background to your data that convinces me that the data are scaled to each basin and that the data were updated recently.
- I don’t know of any country that has good records or consistent data on groundwater extraction (line 146). Similarly, what are ‘representativeness of aquifers and their levels of exploitation’?
-Although Table 2 includes ‘surface water availability’ - Later in the manuscript there is a disclaimer about not knowing whether streams are flowing or dry. Plus disclaimer about associated uncertainty in desired conservation states of medium and deficit (lines 410). These are big disclaimers.
-Conservation Value could use more description. What are gaps and omissions of epicontinental aquatic conservation.
-The discussion of environmental flows is interesting but it wasn’t clear if Mexico has implemented or proposed environmental flows – and if so, I wonder if there really is a comprehensive database to work from? That would be very impressive, but as in other countries, it has been just a dream to have environmental flow guidelines and standards.
-Figure 1 – How does matrix in Fig 1 compare to Equation 1 for each basin? Seems like they are using the same data to come up with different metrics. I am confused and again how are you or the databases consistently coming up with ‘ volumes extracted for productive uses’ across the years?
-I am unclear about the paragraph starting line 236. Was this analysis conducted for this paper? If so, please provide more details. It reads as if you are referencing an existing and published study rather than one that you conducted for this paper. The terms are too vague! How were decisions made country wide for example ‘to identify where water is available in sufficient quantity to ensure environmental flow, which basins are deficient, and the levels of deficiency.’
-Paragraph starting line 261 again describes the value of environmental flows, but how does this paragraph directly tie to what you are calculating? It is justification, but this is not the place for justifying basin conditions.
-in Results – this TABLE 1 (should be Table 3) shares the rankings, but what information went into these rankings? Equation 1? Figure 1? How does this ranking relate to priority to conduct environmental flows? I am confused.
I really would like to like this paper but I need to understand what you are presenting and how you arrived at the determinations that you did.
I suggest you focus the contents of the manuscript and drop sections that aren’t immediately relevant to your analysis, use consistent terminology and definitions for categories, identify clearly the source and spatial and temporal extent of data for each category, present the results in terms of the equation totals, and then summarize into the nice maps and tables that lump equation totals into current condition or potential (such as your A, B, C, D map), and make your summary recommendations as to priority basins.
Comments on the Quality of English Language
The paper is confusing and although some of the confusion might be from translation, I think the original text needs to be revised and clarified by the authors
Author Response
Dear reviewer, thanks for your feedback. Please see attached document with our detailed responses. Kind regards.

Reviewer 2 Report
Comments and Suggestions for Authors
Encourage author to consider greater explanation on Mexico's Norm Methodolgy. Perhaps drawing some information from his earlier paper.
Encourage the author to think about and perhaps comment on how drought cycles affect this evaluation.
The conclusions state the obvious--rivers that are not fully allocated have high potential for reserves and those near full allocation little potential for reserves. That is intuitive.
Author Response
Dear reviewer, thanks for your feedback. Please see below our detailed responses. Kind regards.
Reviewer 2. Encourage author to consider greater explanation on Mexico’s Norm Methodology, perhaps drawing some information from his earlier paper.
Authors response:
We have added the following text in the Introduction to briefly draw on the Mexican Norm methods:
The environmental flow standard sets a three-level hierarchical framework to conduct assessments, from relatively simple and cheap methods (i.e., “look-up tables” and hydrology-based) to more comprehensive and expensive ones (i.e., holistic ecohydrology-, habitat simulation-, expert panel, and research driven-based), and provide water reserve outcomes adjusted to a desired conservation state of basins [11,12]. (p2, lines 63-67)
Greater detail on the Mexican Norm theoretical framework is reasonably provided for the purpose of the manuscript in the Methods section (p7, lines 192-219)
Reviewer 2. Encourage the author to think about and perhaps comment on how drought cycles affect this evaluation.
Authors response:
We have added the following text in the Results & discussion of the Gaps in water reserves for management planning section (3.4. [pp14-15, lines 447-457]):
In general, this result is consistent with the expected as previously reported, both in the method's proof of concept as well as in case studies that include desktop assessments (ecohydrological methods) and others supported by holistic approaches [12,16,17,18,61,62]. However, the reserve volumes should be taken with caution due to the temporal resolution-related uncertainty, particularly in the desired conservation states of "medium" and "deficient", and further considerations should be taken into account.
First, the water availability in the basin is mean annual-based and the extremes are not adequately accounted (i.e. very dry and wet seasonal and yearly conditions). Although at a country and global scale, annual-based water availability and environmental reserves tend to be a regular practice for water planning, there is evidence of inconsistency at greater scales [43,44,45]; drought cycles may affect environmental water reserve previsions provided by reference values. Furthermore, another limitation to be considered is that reference values are applied without differentiation between the type of dominant stream in the basins (e.g., perennial, intermittent, or ephemeral) since it is not possible to determine it from the availability balances [16,17,18]. To increase the certainty in these regards, bottom-up detailed methods such as holistic ecohydrology-based, habitat simulation, expert panel, or research-driven are recommended.
Reviewer 2. The conclusions state the obvious—rivers that are not fully allocated have high potential for reserves and those near full allocation little potential for reserves. That is intuitive.
Authors response:
Yes, you are right, the complete take-away messages that worth to be communicated are that, on one side, (1) the form and geospatial distribution of pressure for water use in Mexico's basins has been advancing at a higher rate compared to the creation of new protected areas (conclusion opening lines, p18 508-510). This is a very important message because the flow regime connectivity protection needs to be grounded on both water and protected areas management. On the other side, (2) the environmental water allocation strategy implemented in Mexico has been successful in terms of the performance assessment adopted in this research, and this has still much more potential to be capitalized (conclusion second paragraph). And, (3) nationwide environmental flow implementation is still possible in 78% of the basins under current water availability conditions based on reference values for water planning designed from strategic bottom-up and top-down experiences (i.e. on-site studies and desktop approaches). This still has great potential in terms of rivers’ connectivity, protected areas, and freshwater-dependent biodiversity conservation (conclusion 2nd-3rd paragraphs). All this is very relevant not only because it is coming from a middle-income, naturally water-scarce country, but also because of its potential replication and escalation.
These outcomes are along the lines of our previous work, which is aligned to and recognized by the state-of-the-art contributions in environmental flow science for freshwater ecosystems conservation (Acreman et al. 2014; Tickner et al 2020; Arthington et al 2023; Piczak et al 2023). In the context of such specialized literature, it makes sense. We aim to bring these messages to this Special Issue too, Methods to Strengthen Protected Area Conservation and Management on the National Level.
Reviewer 3 Report
Comments and Suggestions for Authors
Dear authors,
thank you for an extremely detailed paper, which was a pleasure to read.
The choice of the research topic is also very important, and in the light of climate changes and the future demand for water resources, it will be increasingly significant.
In the work, you only need to eliminate minor technical errors, e.g. lines 113, 143 have undefined references, all years in the list of references must be in bold, etc.
Kind regards
Author Response
Dear reviewer, we are pleased for your appreciation on our manuscript. Thanks for letting us know the minor issues that you pointed out, the paper was thoughtfully revised and corrected. Kind regards.
Reviewer 4 Report
Comments and Suggestions for Authors
I am impressed of the content, idea and quality of the presented manuscript. In my opinion the paper shows novelty, wide research and analyses. Conclusions are proper, clear. Some minor changes are remarked in the attached file. Summarized I appreciate the authors effort and recommend the paper to be published in current form. Great job!

Author Response
Dear reviewer,
Thanks for your impression of our MS, and for pointing out the error on the Table reference in text, it was already amended. About your suggestion on adding the acronym explanation as table footnote, we believe that the correct location is in the Table legend. However, that’s a matter of style and we’ll stick to what the editorial office instructs.
Round 2
Reviewer 1 Report
Comments and Suggestions for Authors
The authors fully addressed by previous questions in their notes to me and they included more information in the text.
Perhaps more of the justification in their notes to me could have been included in the manuscript, but they were working to meet journal guidelines.
Author Response
Dear reviewer,
Many thanks for your feedback. We think that by addressing your comments during the revision our manuscript increased robustness.
Kind regards